# Immune Checkpoint Inhibitor Related Rheumatological Complications: Cooperation between Rheumatologists and Oncologists

**DOI:** 10.3390/ijerph20064926

**Published:** 2023-03-10

**Authors:** Renata Pacholczak-Madej, Joanna Kosałka-Węgiel, Piotr Kuszmiersz, Jerzy W. Mituś, Mirosława Püsküllüoğlu, Aleksandra Grela-Wojewoda, Mariusz Korkosz, Stanisława Bazan-Socha

**Affiliations:** 1Department of Clinical Oncology, The Maria Skłodowska-Curie National Research Institute of Oncology, Kraków Branch, 31-115 Kraków, Poland; 2Department of Anatomy, Jagiellonian University Medical College, 33-332 Kraków, Poland; 3Department of Rheumatology and Immunology, Jagiellonian University Medical Kraków, 30-688 Krakow, Poland; 4Division of Rheumatology and Immunology Clinical, University Hospital, 30-688 Kraków, Poland; 5Department of Surgical Oncology, National Research Institute of Oncology, Kraków Branch, 31-115 Kraków, Poland; 6Department of Internal Medicine, Jagiellonian University Medical College, 30-688 Kraków, Poland

**Keywords:** rheumatic adverse events, immune checkpoint inhibitors, immune-related adverse events, cancer immunotherapy

## Abstract

In cancer, immune checkpoint inhibitors (ICIs) improve patient survival but may lead to severe immune-related adverse events (irAEs). Rheumatic irAEs are a distinct entity that are much more common in a real-life than in clinical trial reports due to their unspecific symptoms and them being a rare cause of hospitalization. This review focuses on an interdisciplinary approach to the management of rheumatic irAEs, including cooperation between oncologists, rheumatologists, and immunologists. We discuss the immunological background of rheumatic irAEs, as well as their unique clinical characteristics, differentiation from other irAEs, and treatment strategies. Importantly, steroids are not the basis of therapy, and nonsteroidal anti-inflammatory drugs should be administered in the front line with other antirheumatic agents. We also address whether patients with pre-existing rheumatic autoimmune diseases can receive ICIs and how antirheumatic agents can interfere with ICIs. Interestingly, there is a preclinical rationale for combining ICIs with immunosuppressants, particularly tumor necrosis factor α and interleukin 6 inhibitors. Regardless of the data, the mainstay in managing irAEs is interdisciplinary cooperation between oncologists and other medical specialties.

## 1. Introduction: The Role of Immune Checkpoint Inhibitors in Oncology

Immune checkpoint inhibitors (ICIs) have demonstrated efficacy in many different types of malignancy, improving the prognosis of cancer patients. This treatment strategy has been a significant breakthrough in oncology in recent years. What is unique is that once the patient responds, remissions are long-lasting and, depending on the type of cancer, can reach even 50–60% [1,2]. Primary ICIs have been used in advanced-stage cancers, but their position in treatment strategies has shifted to adjuvant, neoadjuvant, and consolidation settings.

Currently, the ICIs approved by the European Medicines Agency (EMA) and the U.S. Food and Drug Administration (FDA) are those directed against cytotoxic T-lymphocyte antigen 4 (CTLA-4), programmed cell death receptor-1 (PD-1) or its ligand (PD-L1), and lymphocyte activation gene-3 (LAG-3). Other novel molecules, such as anti-T-cell immunoglobulin and mucin-domain containing-3 (TIM-3), anti-T-cell immunoglobulin and immunoreceptor tyrosine-based inhibitory motif domain (TIGIT), and anti-V-domain Ig suppressor of T-cell activation (VISTA), are being evaluated in clinical trials.

Most patients treated with ICIs (82% in monotherapy and >90% in combination) will develop some autoimmune reactions. Immune-related adverse events (irAEs) can affect any organ or system. Fortunately, the most common irAEs, such as fatigue, diarrhea, pruritus, or nausea, are mild and easy to treat (e.g., the Checkmate 067 study [3]).

Rheumatic complications with ICIs involve a specific group of irAEs. They are chronic and usually last, although immunotherapy cessation. Furthermore, they can occur even years after treatment cessation; their therapy is also challenging [4]. In the study of Ghisoni et al. [5], the onset of different rheumatic irAEs was reported after a median of 71 days (28 days in melanoma and 167 days in lung cancer) and lasted for a median of 169 days (85 days in melanoma and 236 days in lung cancer) after a first dose of ICIs. Interestingly, the frequency of rheumatic irAEs seems to be underestimated since their symptoms are commonly unspecific and rarely lead to hospitalization or fatality [6,7]. Therefore, they are more frequently observed in real-life than in reports from clinical trials with highly selected populations. Among rheumatic irAEs, the most common are arthralgia, myalgia, arthritis, vasculitis, and myositis. In contrast, the others, including sarcoidosis, polymyalgia-rheumatica-like syndrome, systemic sclerosis, lupus erythematosus, sicca syndrome, and unspecific bone pain, are much less frequent [2,8].

This review aimed to characterize rheumatic irAEs in terms of their diagnostic and treatment strategies as a result of discussion between oncologists, rheumatologists, and immunologists. We also report whether patients with pre-existing rheumatic disease can receive ICIs and how antirheumatic agents may interfere with those medications. Eventually, we present the theoretical rationale for combining ICIs with immunosuppressants in oncology.

## 2. Methods

In order to address this objective, we searched the PubMed database using key words, such as rheumatic adverse events, immune-related adverse events, immunotherapy, rheumatic diseases, arthritis, and myositis. Afterwards, we checked the reference lists of the included studies for further references. Additionally, we referred to the international guidelines of the European Society for Medical Oncology [9] and the European Alliance of Associations for Rheumatology (EULAR) [10].

## 3. Suppressor Modulation of T Cell and Antigen-Presenting Cell Cooperation by Immune Checkpoint Inhibitors

The main goal of T-cell activation is to generate a fraction of lymphocytes from naïve T cells, which differentiate into precursor T cells [11,12]. These precursor cells ultimately develop into central memory (T_CM_), effector memory (T_EM_), and tissue-resident memory (T_RM_) T cells [11,12,13,14]. The proliferation and differentiation of naïve T cells require providing signals primarily by molecules localized on antigen-presenting cells (APCs) [15,16]. The first signal is related to antigen recognition by the T-cell receptor (TCR) in the context of the major histocompatibility complex (MHC) on APC. The second signal refers to co-signaling molecules, i.e., cell-surface glycoproteins, that can directly modulate and fine-tune the TCR signal [17]. Based on their functional outcome, co-signaling molecules can be divided into co-stimulators and co-inhibitors, which promote or suppress T cell activation [15,16,17]. Without sufficient TCR signaling, co-signaling molecules lose or change their function aberrantly [17,18]. Similarly, without co-stimulation, T cells, even encountering specific antigens, fail to respond, enter apoptosis, or remain anergic [11,17,18].

In Figure 1, we present an immunological synapse with all the main stimulatory and inhibitory molecules. The best-characterized co-stimulatory pathway in T cell activation involves the surface cluster of differentiation (CD) 28, which binds the co-stimulatory molecules B7-1 (CD80) and B7-2 (CD86) expressed on activated APCs [11,16,17]. The expression of CD80/CD86 co-stimulators is regulated and ensures that T-cell response is appropriately initiated [11]. In addition, activated T cells expresses the surface CD40 ligand (CD40L), which binds to CD40 expressed on APC and delivers signals that enhance the expression of B7 on the APC [11]. This feedback loop serves to amplify T-lymphocyte responses [11]. Another co-stimulatory molecule on T cells is the so-called “inducible T-cell co-stimulator” (ICOS, CD278) [11,16,17]. Its ligand (ICOS-L, CD275) on APC plays a crucial role in T-cell-dependent antibody responses, mainly in the germinal center (GC) reaction, which takes place in secondary lymphoid organs, such as lymph nodes [11]. It needs to develop and activate follicular helper T cells, which provide essential activating signals to B cells in GCs [11].

However, the outcome of T-cell activation depends on a balance between the engagement of activating and inhibitory receptors of the CD28 family members [11,16,17]. The main co-inhibitor molecule from that group is the cytotoxic T-cell antigen 4 (CTLA-4) [11,16,17]. It likely acts by competing with CD28 for B7 and winning out because it has a higher affinity for B7 [16]. The capture of B7 by CTLA-4 into a zipper-like structure closes down B7–CD28 signaling, thereby inhibiting the production of interleukin-2 (IL-2) and T-cell proliferation [16]. Additionally, evidence for the inhibitory function of CTLA-4 includes the enhancement of T-cell responses against foreign and self-antigens by monoclonal antibodies that block the interaction between CTLA-4 and CD80/CD86 [16,17]. The subsequent co-inhibitor molecules, this time from the B7 family, are PD-1 (CD279) [11,16,17] and B- and T-lymphocyte attenuator (BTLA, CD272), which are found only on activated cells [17]. BTLA-deficient T cells have increased responses to antigen stimulation in vitro [19].

Other T-cell surface molecules, including CD2 and integrins, also deliver co-stimulatory signals in vitro. Still, their physiological role in promoting T-cell activation is less clear than that reported for the CD28 family [11]. In addition, recent studies have identified several new immune checkpoints targets, such as LAG-3, TIM-3, TIGIT, and VISTA [7,20,21]. However, investigations of these molecules in preclinical and/or clinical studies are promising yet preliminary [20,21].

Potential therapeutic strategies in modifying T-cell and APC cooperation are provided in Figure 1. They are used in oncologic, immunologic, and rheumatologic treatment. ICIs are presented in a solid line frame with division into agents approved in clinics (in green) and those currently under investigation (in red).

In oncology, the main goal is the blockage of inhibitory receptors that, as a result, activates T cells and induces the immunologic response against malignant cells. Conversely, in autoimmunity, the target is the opposite, i.e., activating inhibitory receptors or suppressing activating receptors leading to a decrease T-cell activity (Figure 1). However, inhibitory receptors also play a role in chronic inflammatory diseases [22], and this general assumption has found justification in laboratory and clinical studies. For example, in the synovium of patients with rheumatoid arthritis, PD-1/PD-L1 is up-regulated and influences T-cell response [23], and, in another study, the soluble PD-1 concentration was correlated with the titer of rheumatoid factor (RF) [24]. Furthermore, knocked-out mice deficient in one of the inhibitory receptors were found to be susceptible to developing autoimmune diseases [22]. In the therapeutic approach, abatacept, registered in the treatment of rheumatoid, juvenile idiopathic, and psoriatic arthritis, acts by blocking the co-signaling between CD28 and CD80/86, which is a stimulating signal for T cells. Therefore, this agent can be considered as an opposition to a registered anti-CTLA-4 antibody—ipilimumab (Figure 1) [25].

## 4. Risk Factors for-Rheumatic Adverse Events Related to Immune Checkpoint Inhibitors Are Unknown

Unfortunately, there are no specific clinical or molecular markers to predict the risk of the occurrence of rheumatic irAEs, although many studies have aimed to address this issue. For example, Cunningham-Bussel et al. [26], in their retrospective report based on medical records of more than 8000 patients, demonstrated that patients (1) with tumors considered as ‘inflammatory’ with a high mutation burden (for melanoma: odds ratio (OR), confidence interval (CI): 2.54–6.51, and for genitourinary cancers: OR 2.22, CI: 1.39–3.54), (2) treated with combination immunotherapy (OR 2.35, CI: 1.48–3.74), (3) with a medical history of any autoimmune disease (OR 2.04, CI: 1.45–2.85), and (4) who recently used steroids from any cause (OR 2.13, CI: 1.51–2.98), were found to be more prone to develop rheumatic irAEs during ICIs oncologic treatment. The correlation between baseline autoimmune disease and the risk of rheumatic irAEs is depicted in Section 7.2.

Class I and II human leukocyte antigen (HLA) alleles are established genetic risk factors for a variety of autoimmune diseases [27]. In the case of rheumatoid arthritis (RA), the HLA-DRB1, HLA-A, HLA-B, and HLA-DPB1 genes constitute genetic susceptibility for the development of the disease [28]. This finding became the rationale for the pilot study by Cappelli et al. [29]. The HLA alleles found in the classic form of RA were present in 61.5% of patients who developed ICIs arthritis. Additionally, the authors reported that HLA B * 52: 01 and C * 12: 02 may be associated with ICIs arthritis. HLA class I genotypes may also influence the response to ICIs [30]. This issue is an intriguing one that deserves further research.

## 5. General Principles and Initial Approach to Rheumatic Adverse Events in Patients on Immune Checkpoint Inhibitors

A multidisciplinary team approach with at least one oncologist and rheumatologist is crucial for proper decision-making. The rheumatologist’s contribution refers to the knowledge of autoimmune disease symptomatology and the management of rheumatic irAEs, particularly using immunosuppressive agents and biologics in those with steroid-refractory outcomes. Finally, the patient’s preference must be engaged in the final treatment plan. In a differential diagnosis, it is necessary to consider metastases, paraneoplastic syndromes, primary rheumatic disease, and other etiologies such as infections, thromboembolism, and endocrinopathies [10]. Diagnosis is even more difficult than in the ‘standard’ approach due to the common absence of autoantibodies and the normal concentration of acute-phase reactants [31,32]. Additionally, an X-ray and ultrasound of affected joints and advanced imaging methods are recommended to differentiate between the entities mentioned below [9,10]. Tissue biopsy is not mandatory if it does not change the treatment strategy. However, the EULAR recommends a histopathological confirmation of vasculitis, sarcoidosis, and myositis, but it should not delay the initiation of treatment [10]. After establishing a proper diagnosis, the disease severity should be classified according to the Common Terminology Criteria for Adverse Events (CTCAE version 5.0 [33]), the standard for oncologists, but this is usually not applied by rheumatologists. Therefore, classification should not be the main paradigm in determining whether a patient needs to be referred to a rheumatologist. Ideally, a suspicion of rheumatic irAEs and no improvement after preliminary therapy with nonsteroidal anti-inflammatory drugs indicates a need for referral to a rheumatologist, even in case of mild AE, such as grade 1 (G1). Such an approach facilitates diagnosis and treatment initiation, which is different from other irAEs [9]. Rheumatologists have created their criteria for describing AEs in clinical trials (Table 1). This classification includes symptom duration, lifestyle changes, and treatment effects [34]; however, it is not widely used and is unknown by oncologists.

Similar to other irAEs, it is most important to find a balance between treating the tumor through immunotherapy and treating irAEs through immunosuppression. The treatment of rheumatic irAEs is similar to primary rheumatic diseases. It is characterized in detail in the next section and in Table 2. Patients who experience AEs G1–G2 should start therapy with nonsteroidal anti-inflammatory drugs and/or other analgesics. In the cases of arthritis of a limited number of joints, intraarticular steroid administration might be the best option. Systemic steroids are used in more severe cases (e.g., AE G3) or those resistant to the prior regimen [10]. In patients requiring steroid-sparing or those with steroid-refractory rheumatic irAEs, EULAR recommends the application of disease-modifying antirheumatic drugs (DMARDs) such as methotrexate and sulfasalazine in first-line treatment or biologics (bDMARDs), including tocilizumab, infliximab, and etanercept in particularly resistant cases. However, they must be administered for a long time, as rheumatic irAEs are more chronic and persistent for a long time [35]. On the contrary, in the treatment of colitis, one dose of infliximab is sufficient for disease remission [9]. The debate on treatment strategies has also raised different views, suggesting that DMARDs could be used in the front line to protect patients from exposure to steroids [6].

## 6. Clinical Characteristics and Management of the Rheumatic Adverse Events Related to Immune Checkpoint Inhibitors 

### 6.1. Inflammatory Arthritis

Arthritis is the most commonly observed rheumatic complication in oncologic patients treated with ICIs, with a reported incidence of 1.5–22% in retrospective and prospective real-life data [2]. Furthermore, in the study of Cappeli et al. [36], 86% of the affected patients had related symptoms months after the discontinuation of immunotherapy.

In the clinical presentation, ICIs arthritis can present forms similar to rheumatoid arthritis, psoriatic arthritis, or reactive arthritis with sterile urethritis and conjunctivitis, as well as in the form of arthritis with accompanying colitis or spondylarthritis. Research suggests that the pattern of arthritis may differ depending on the exposed ICIs. For example, patients treated with anti-PD-1/PD-L1 monotherapy are at an increased risk of developing inflammation of the small joints. At the same time, knee involvement and reactive-arthritis-like symptoms are more common when combined with anti-CTLA-4 [36].

In the clinical evaluation of patients with suspected ICIs arthritis, differential diagnosis to the corresponding rheumatic disease must be taken into account. Patients with rheumatoid-arthritis-like polyarthritis are much less likely to have RF or anti-cyclic citrullinated peptide antibodies (anti-CCP) [31]. Furthermore, there is no female prevalence, and erosions may be present early in the disease, as well as tendon involvement. In the case of forms similar to psoriatic arthritis or spondylarthritis, the presence of human leukocyte antigen (HLA) B27 is not often documented, as is the presence of accompanying psoriatic skin lesions.

For this reason, in the diagnosis of patients with ICIs arthritis, the determination of inflammatory markers and autoantibodies may be helpful, but it is not the basis for the diagnosis. Joint radiographs are recommended as a part of the routine assessment, primarily due to the early onset of erosions. Ultrasound techniques and magnetic resonance imaging of selected joints may also be valuable [37].

Treatment of patients with ICIs arthritis should follow the patient–oncologist–rheumatologist axis and consider rheumatic symptoms and oncologic goals. The management is based on graded CTCAE algorithms, which may not fully reflect the severity spectrum of rheumatological manifestations. A summary of therapeutic recommendations is presented in Table 2, taking into account the pattern of arthritis [38,39]. Importantly, the steroid dose used in ICIs arthritis therapy needs to be low (optimum less than 10 mg of prednisone/day), avoiding high doses in pulses, in order to minimize their unfavorable effects on the cellular antitumor immunologic response [40] (described in detail in the Section 7.3).

### 6.2. Inflammatory Myopathies

Myositis related to ICIs is a relatively rare complication. Its prevalence is approximately 1% among all patients treated with these medications. However, among all irAEs, it is associated with the highest risk of lethal outcomes. In addition, the relatively short time (the median of about 25 days) from the beginning of the use of ICIs to the first myositis symptoms needs to be emphasized [41].

The clinical presentation in most cases resembles polymyositis, with elevated markers of muscle damage, such as creatine kinase and myoglobin. In contrast to primary polymyositis, myositis-specific autoantibodies are frequently not detected. On the other hand, when present, the most prevalent are anti-PM/Scl, anti-TIF1 gamma, and anti-PL-7 [42,43]. Pathological skin changes and the typical pattern of dermatomyositis are described very rarely. Muscle biopsy may be helpful in questionable cases because histopathology often shows characteristic lymphocytic infiltrations with CD8 and CD4 T lymphocytes and myofiber necrosis [44].

It is important to distinguish between the myositis-specified antibodies mentioned above from myositis-associated ones (anti-Ro/SSA, anti-DNAPK, anti-PM-Scl, and anti-Scl70), which characterize overlapping syndromes such as systemic sclerosis and anti-synthetase syndrome (e.g., antibodies against histidyl-transfer-RNA-synthetase (anti Jo-1)) [45]. The former is extremely rare, and there are only two case reports of patients with limited and diffuse skin involvement and an absence of autoantibodies; thus, the diagnosis was established based on clinical symptoms (fatigue, muscle weakness, skin tightness, etc.) [46]. The latter was reported in one patient after eight courses of nivolumab with positive anti-threonyl-tRNA synthetase antibodies (anti-PL-7) who presented with muscular weakness and respiratory failure during interstitial lung disease [47]. In each case, the disease must be treated as a corresponding rheumatic entity.

It should also be noted that in ICIs myositis, myocarditis and myasthenia gravis may coincide. Their presence significantly affects the prognosis and is associated with a higher risk of death. Therefore, such patients should be treated more aggressively with high doses of steroids, and in the majority of them, the continuation of ICIs, even after symptoms withdrawal, is not recommended.

### 6.3. Polymyalgia Rheumatica (PMR)

Data from the case series indicate that almost 20% of patients with rheumatic irAEs present with a polymyalgia-rheumatica-like syndrome. As in the primary form, this complication is typical for patients over 50 years of age [48]. In addition, the clinical characteristics of giant cell arteritis may also be present with pathological findings in a biopsy, similar to the primary syndrome [7].

However, in ICIs-related polymyalgia-like syndrome, inflammatory markers (erythrocyte sedimentation rate, C-reactive protein) are usually lower than in the primary rheumatic disease. Furthermore, remission might not be obtained even after high steroid doses.

In the therapeutic approach, in most cases, steroids are used for several weeks with a gradual dose reduction [49]. According to the recommendations of the National Comprehensive Cancer Network (NCCN), in mild cases with shoulder girdle involvement, the use of local steroid injections may be considered to reduce the systemic side effects of steroids [50].

### 6.4. Sicca Syndrome and Ocular Disorders

The clinical presentation of sicca syndrome caused by ICIs resembles Sjogren’s syndrome, presenting as symptoms of dry eyes and mouth. They typically develop within the first 3 months of therapy. Clinical case reports indicate that xerostomia is more common than xeropthalmia. Furthermore, the vast majority of patients do not show the presence of typical anti-Ro and anti-La autoantibodies [41]. In some cases, the histopathological findings of the labial salivary gland document a disseminated infiltration of T cells. In managing this syndrome, in addition to typical symptomatic treatment, low-dose steroids play an important role [51].

### 6.5. Others: Vasculitis, Sarcoidosis, and Lupus Erythematosus

During ICIs therapy, patients may experience other rheumatic irAEs resembling several different autoimmune syndromes.

Cases of sarcoidosis and sarcoid-like reactions have been reported in the literature, most often diagnosed with new hilar lymphadenopathy and pulmonary nodules [52]. In the clinical presentation, erythematous skin changes, cough, and arthralgia were also described, while ocular and neurological manifestations were sporadic.

Vasculitides of all vascular sizes have also been reported during ICIs treatment. For example, typical glomerulonephritis with lung damage and the presence of anti-neutrophil cytoplasmic antibodies (ANCA) has been reported in a patient treated with anti-PD-1 antibodies [53]. Polyneuropathy, a hallmark of vasculitis, must be differentiated from chemotherapy-induced polyneuropathy, paraneoplastic syndromes, or direct infiltration by the tumor [54]. Typical characteristics of vasculitis neuropathy include sensory/sensorimotor involvement with pain and relapsing course, asymmetry and/or multifocality, and lower limbs and distal predominance. In addition, electromyography demonstrates evidence of axonal, sensorimotor, or sensory neuropathy. A nerve biopsy is not necessary [54,55].

Although there have been some reports of lupus nephritis associated with ICIs [56], full-symptomatic lupus erythematosus has not yet been reported.

In the case of those rare rheumatic irAEs, universal diagnostic and therapeutic approaches have not yet been established; thus, close cooperation between the oncologist and rheumatologist is of critical importance. In severe and refractory cases, bDMARDs and tyrosine kinase inhibitors can be considered.

**Table 2 ijerph-20-04926-t002:** Summary of rheumatic complications after immune checkpoint inhibitors with treatment strategies depending on a grade according to the Common Terminology Criteria for Adverse Events version 5.0.

Rheumatic Immune-Related Adverse Events	Signs and Symptoms	Confirmatory Tests	Treatment
**Rheumatoid-arthritis-like**	Symmetric peripheral polyarticular phenotype. Joint pain accompanied by swelling, morning stiffness with improvement after anti-inflammatory agents.	- Complete number of tender and swollen joints; - Spine examination; - X-ray or US joint examination; - Blood panel including RF, anti-CCP, ESR, and CRP; - Consider MRI of affected joints.	G1: First line: NSAIDs/intra-articular corticosteroid, prednisone up to 20 mg/d for 4 weeks. Second line: DMARDs (hydroxychloroquine, sulfasalazine, methotrexate, and leflunomide). *Continuation of ICIs* G2: First line: NSAIDs/intra-articular corticosteroid, prednisone up to 0.5 mg/kg/d for 2–3 weeks with tapering. Second line: csDMARDs (sulfasalazine, methotrexate, and leflunomide) or bDMARDs (anti-TNFα and anti-IL-6). *Temporal discontinuation of ICIs until G0/G1* G3–4: First line: NSAIDs/intra-articular corticosteroid, prednisone up to 1 mg/kg/d for 2–3 weeks with tapering. Second line: csDMARDs (methotrexate in doses ≥15 mg weekly) with bDMARDs (anti-TNFα and anti-IL-6). *ICIs cessation*
**Psoriatic-arthritis-like**	Mono/oligoarticular phenotype. Joint pain accompanied by swelling, morning stiffness with improvement after anti-inflammatory agents.	- Complete number of tender and swollen joints; - Spine examination; - X-ray or US joint examination; - Blood panel including RF, anti-CCP, ESR, and CRP; - Consider MRI of affected joints and tendons; - Dermatologist consultation and skin assessment (e.g., PASI).	G1: NSAIDs/intra-articular corticosteroid, consider methotrexate. *Continuation of ICIs* G2: NSAIDs/intra-articular corticosteroids with or without csDMARDs (sulfasalazine/methotrexate and azathioprine) or bDMARDs (anti-TNFα, antiIL-17, anti-IL-12/23, and JAK inhibitors). *Temporal discontinuation of ICIs until G0/G1* G3–4: NSAIDs/intra-articular corticosteroid with csDMARDs (sulfasalazine/methotrexate and Azathioprine) or bDMARDs (anti-TNFα,anti-IL-17, anti-IL-12/23, andJAK inhibitors). *ICIs cessation*
**Spondylarthritis -like**	Peripheral/axial spondyloarthropathy with or without a colitis phenotype. Joint pain accompanied by swelling, morning stiffness with improvement after anti-inflammatory agents.	- Complete number of tender and swollen joints; - Spine examination; - X-ray or US joint examination; - Blood panel including RF, anti-CCP, ESR, and CRP; - Consider MRI of affected joints and tendons; - Blood panel including HLA-B27; - Endoscopy when colitis suspected.	G1: NSAIDs/sacroiliac corticosteroid. *Continuation of ICIs* G2: NSAIDs/sacroiliac corticosteroid with or without bDMARDs (anti-TNF, anti-IL-12/23, and JAKi; avoid anti-IL-17 when colitis). *Temporal discontinuation of ICIs until G0/G1* G3–4: NSAIDs/sacroiliac corticosteroid with bDMARDs (TNFi, anti-IL-12/23, and JAKi; avoid anti-IL-17 when colitis). *ICIs cessation*
**Inflammatory myopathies**	Muscle weakness and pain. Typically, weakness of the proximal limbs. Possible presence of bulbar symptoms (dysphagia, dyspnea, and diplopia), myocarditis, and interstitial lung disease. Typical for dermatomyositis skin lesions to occur.	- Muscle strength examination; - Skin evaluation; - Evaluation of vision and ocular muscles; - Blood panel including CK, AST, ALT, LDH, aldolase, troponin, NT-proBNP, ESR, CRP, ANA (with myositis panel, e.g., Jo1,Mi-2,SRP, etc.), and AChR antibodies; - Electrocardiogram or Echocardiogram; - Consider electromyography, MRI, and muscle biopsy.	G1: First line: NSAIDs with or without prednisone 0.5 mg/kg/d. Second line: DMARDs (mycophenolate mofetil, methotrexate, and Azathioprine). *Continuation of ICIs* G2: First line: NSAIDs with or without prednisone 1 mg/kg/d Second line: DMARDs (mycophenolate mofetil, methotrexate, and azathioprine). *Temporal discontinuation of ICIs until G0/G1* G3: High-dose systemic steroids (1–2 mg/kg/d) with or without immunosuppressive intravenous immune globulin dose. Second line: rituximab, tacrolimus, and abatacept *ICIs cessation* G4: Pulse-dose systemic steroids with or without immunosuppressive intravenous immune globulin dose and plasmapheresis. Second line: rituximab, tacrolimus, and abatacept. *ICIs cessation*
**Polymyalgia rheumatica**	Symmetrical pain and stiffness of the proximal upper/lower extremities without muscle injury. In rare cases associated with large vessel vasculitis.	- Joints and skin examination; - Blood panel including ESR, CRP, ANA, RF, anti-CCP, and CK; - Vision assessment; - Consider the US examination of the temporal artery.	G1: First line: prednisone up to 20 mg/d for 6 weeks/intra-articular corticosteroid. Second line: methotrexate. *Continuation of ICIs* G2: First line: prednisone up to 30 mg/d for 6 weeks/intra-articular corticosteroids. Second line: methotrexate and anti-IL-6. *Temporal discontinuation of ICIs until G0/G1* G3–4: First line: pulse-dose systemic steroids. Second line: anti-IL-6 biologics. *ICIs cessation*
**Sicca syndrome and ocular disorders**	Sjogren-like syndrome. Eyes/mouth/genital area dryness. Joints and muscle pain possible. Uveitis, peripheral ulcerative keratitis, and other forms of ocular inflammation.	- Examination of the eyes and mouth; - Labial salivary gland biopsy; - Blood panel including ESR, CRP, ANA, RF, anti-CCP, and CK; - Consider the US examination of salivary glands.	G1: First line: saliva substitute/artificial tears. Second line: pilocarpine or other sialagogues and hydroxychloroquine. *Continuation of ICIs* G2–4: Prednisone up to 40 mg/d for 4 weeks. *Temporal discontinuation of ICIs until G0/G1 or ICIs cessation*
**Vasculitis**	Skin lesions: purpura, ulcers, bloody nasal discharge, edema, dyspnea, cough, hemoptysis, hematuria, proteinuria, and polyneuropathy.	- Physical examination with rheumatologic assessment; - Imaging of the affected area: CT, MRI, and US; - Lesions biopsy; - HRCT of the head, ears, and chest; - Blood panel including: ANCA, ESR, and CRP; urine test with sediment.	G1–G2: Prednisone up to 1–2 mg/kg/d or pulses and hydroxychloroquine for skin vasculitis). Second line: cyclophosphamide and methotrexate. *Temporal discontinuation of ICIs until G0/G1* G3-G4: Rituximab or plasma exchange. *ICIs cessation*
**Systemic sclerosis ***	Skin thickening, sclerodactylitis, phalangeal ulcers, Raynaud’s phenomenon, calcinosis, dyspnea, and cough.	- Physical examination with rheumatologic assessment; - Imaging of the affected area: CT, MRI, and US; - Skin assessment; - Capillaroscopy; - Endoscopy; - HRCT of chest; - Blood panel including: ANA, ESR, and CRP; urine test with sediment.	G2–4: Prednisone up to 1 mg/kg/d with or without hydroxychloroquine. Second line: mycophenolate mofetil, azathioprine, and methotrexate (cyclophosphamide in interstitial lung disease). *Temporal discontinuation of ICIs until G0/G1 or ICIs cessation*
**Sarcoidosis ***	Arthralgia, uveitis, cough, dyspnea, and lymphadenopathy.	- Physical examination with rheumatologic assessment; - Imaging of the affected area: CT, MRI, and US; - HRCT of the chest; - Bronchoscopy; - Blood panel that includes: RF, anti-CCP ESR, CRP, and sCD25+.	G2–4: Prednisone up to 1 mg/kg/d. Taper steroids over 2–4 months. Second line: methotrexate, azathioprine, and mycophenolate mofetil. *Temporal discontinuation of ICIs until G0/G1 or ICIs cessation*

Table 2 references: [8,10,57,58,59,60,61] *—lack of specific guidelines for ICIs-related systemic sclerosis and sarcoidosis. Treatment recommendations are presented as for the classic forms of these diseases. Abbreviations (in alphabetic order): AChR—acetylcholinesterase antibodies, ALT—alanine aminotransferase, ANCA—antineutrophil cytoplasmic antibodies, ANA—antinuclear antibodies, anti-CCP—anti–cyclic citrullinated peptide antibodies, AST—aspartate aminotransferase, bDMARDs—biologic disease-modifying antirheumatic drugs, csDMARDs—conventional synthetic disease-modifying antirheumatic drugs, CK—creatine kinase, CRP—C-reactive protein, CSI—corticosteroid injections, CT—computer tomography, DMARDs—disease-modifying antirheumatic drugs, ESR—erythrocyte sedimentation rate, G-grade, HLA—human leukocyte antigen, HRCT—high-resolution computed tomography, ICIs—immune checkpoint inhibitors, IL—interleukin, JAKi—Janus kinase inhibitors, LDH—lactate dehydrogenase, MRI—magnetic resonance imaging, NSAIDs—nonsteroidal anti-inflammatory drugs, NT-proBNP—N-terminal pro-brain natriuretic peptide, PASI—Psoriasis Area and Severity Index, RF—rheumatoid factor, sCD—soluble cluster of differentiation, TNFi—tumor necrosis factors alfa inhibitors, US—ultrasound imaging.

Additionally, symptomatic and supportive treatment may be introduced, e.g., as follows:Raynaud’s phenomenon—avoidance of cold exposure and vasoconstricting drugs, smoking cessation, calcium channel blockers/phosphodiesterase type 5 inhibitors/topical nitrate/angiotensin II receptor blockers [60];Arthralgia—non-opioid analgesics such as nonsteroidal anti-inflammatory drugs or acetaminophen [59]; physical therapy—exercise is generally a safe option for all cancer patients, but for those with active disease, some forms of physical modalities are contraindicated, including heat, ultrasound, transcutaneous electrical nerve stimulation, functional electrical stimulation, low-level light laser, and manual therapy (for more information see [61]);Kidney disease—blood pressure control using angiotensin-converting enzyme inhibitors, avoiding values lower than 120/80 mmHG [62];Neuropathy—rehabilitation and orthoses, pain control using amitriptyline/nortriptyline, duloxetine, gabapentin, or pregabalin with avoidance of opioids [63];Osteoporosis prevention—for patients receiving any dose of steroids for ≥3 months, a total calcium intake of 1000–1200 mg/day and vitamin D intake of 600–2000 international units/day through either diet and/or supplements maintained with occasional vitamin D and calcium serum concentration controlling [64];Prevention of opportunistic infections—prophylactic vaccinations (optimal before immunosuppressive therapy begins) for patients receiving high-dose steroids and/or other immunosuppressive agents consider pneumocystis pneumonia prophylaxis using trimethoprim-sulfamethoxazole [54].

## 7. Oncologic Immunotherapy with Checkpoint Inhibitors in Patients with Pre-Existing Rheumatic Diseases Seem to Be Safe, Albeit Needing Vigilance

ICIsadministration in patients with pre-existing rheumatic diseases raises three baseline questions: (1) what is the risk of worsening the pre-existing condition? (5.1); (2) is the risk of irAEs higher than in the general population? (5.2); do drugs used for rheumatic disease influence the effectiveness of ICIs? (5.3).

### 7.1. Pre-Existing Rheumatic Diseases Might Worsen during Immune Checkpoint Inhibitor Therapy

ICIs data regarding cancer patients with pre-existing rheumatic conditions come mainly from real-world studies [65,66], case series [67], or systematic reviews [68], since that patient population has not been enrolled in clinical trials [69]. Table 3 summarizes their results; however, they all focused on autoimmune disorders rather than specific rheumatic diseases.

The general precaution for these patients is listed below.

Rheumatologists should closely monitor patients with pre-existing rheumatic diseases since properly controlling the disease is crucial before administering ICIs. As expected, patients with rheumatism experienced flares more often than those with non-rheumatic autoimmune disorders (40% vs. 10%; *p* = 0.01) [70]. Hence, approximately 25–40% of those with rheumatic diseases might experience worsening upon receiving ICIs [66,69,71,72]. Fortunately, flares are usually mild and manageable without needing immunotherapy withdrawal [68].

Generally, the mutual administration of ICIs and antirheumatic drugs, such as hydroxychloroquine and sulfasalazine, is safe and does not diminish the efficacy of immunotherapy [25]. On the other hand, biologics (bDMARDs) have not been evaluated in such a setting [25]. Nonetheless, it has been suggested that the replacement of non-selective immunosuppressants, including steroids, mycophenolate mofetil, cyclophosphamide, and methotrexate, with precisely acting rituximab (anti-CD20), vedolizumab (anti-α_4_β_7_ integrin), tocilizumab (anti-IL-6), or anti-IL-12/23 blockers might be a better option in the management of cancer patients. However, such an approach has not been validated in clinical trials and is based only on a theoretical basis [73]. Nevertheless, this strategy may lead to effective therapy using ICIs, reducing the risk of rheumatic disease flares [74,75].

In conclusion, ICIs seem to be a safe option for patients with pre-existing rheumatic diseases; however, at enrollment and follow-up, rheumatologists and oncologists must carefully evaluate them for the potential risk–benefit ratio [76].

**Table 3 ijerph-20-04926-t003:** Summary of studies regarding the population of cancer patients with pre-existing rheumatic conditions receiving immune checkpoint inhibitors.

Population	Year	Type of Study	No of Patients with Autoimmunity	No of Patients with Rheumatic Diseases	Outcome *	Ref.
Italian patients with pre-existing autoimmunity and advanced cancer receiving anti-PD-1 mAbs.	2019	Retrospective observationalMulticenter	85	10	- irAEs of any grade were significantly higher in the population with autoimmunity (both active and inactive) in comparison to the general population; - There were no differences in terms of toxicities G3 or G4 as per CTCAE or in response to the treatment; - Toxicities of any grade were experienced by 65.9%, and toxicities G3 or G4 were experienced by 9.4% of the population with autoimmune diseases; - A total of 47.1% of patients experienced a flare of their disease.	[66]
French population with pre-existing autoimmunity and cancer treaded with ICIs.	2019	Retrospective observationalmulticenter	112	20	- A total of 47% of patients had flares of pre-existing conditions; - A total of 43% of patients experienced irAEs; - A total of 43% of patients with a flare or irAEs required immunosuppressants, and 21% discontinued—patients receiving immunosuppressants at treatment initiation and these experiencing irAEs or flare of autoimmune disorder had worse PFS;	[69]
Patients with autoimmune disorders and cancer receiving ICIs (49 publications).	2018	Systematic review	123	68	- A total of 75% of patients had a worsened autoimmune disease or/and irAEs; - There were no differences in irAEs between patients with an active or inactive condition; - Immunosuppressants at ICI initiation resulted in fewer irAEs; - Disease exacerbation and irAEs were manageable with steroids, with fewer than 20% of patients requiring other immunosuppressive agents.	[68]
Cancer patients with pre-existing rheumatic conditions receiving ICIs.	2018	Single-center case series	N/A	16	- irAEs occurred in 38% of patients; - Patients with irAEs were treated with glucocorticoids and did not continue ICIs.	[67]
Greek cancer patients with pre-existing autoimmune conditions receiving ICIs.	2022	Retrospective observationalmulticenter	123	54	- A total of 25.2% of patients experienced flares of pre-existing conditions; - A total of 35% of patients experienced new irAEs of any grade (including thyroiditis, dermatitis, or colitis); - A total of 8.9% of patients discontinued ICIs due to toxicity; - irAEs occurrence was correlated with longer PFS; - Steroids treatment at ICIs initiation was correlated with poorer PFS.	[65]
Cancer patients with pre-existing rheumatic conditions receiving ICIs.	2022	Single-center case series	N/A	45	- A total of 29% of patients experienced flares of pre-existing conditions; - A total of 44% of patients experienced new irAEs of any grade; - Higher risk of a flare was associated with rheumatoid arthritis in comparison to other diseases, but there were no toxicities > G2 in the rheumatoid arthritis population; - Anti-inflammatory treatment at ICIs initiation was not associated with a lower risk of flare or irAEs or with treatment outcome.	[77]
Cancer patients with pre-existing inflammatory or autoimmune conditions receiving anti-PD-1 and registered in REISAMIC registry.	2018	Prospective, multicenter national registry (real-world database)	45	11	- A total of 24.4% of patients experienced a flare of pre-existing condition; - A total of 20% of patients experienced new irAEs of any grade; - ICIs treatment was stopped in five patients; - OS and ORR was similar in group with pre-existing conditions and in the general population; - Patients with autoimmune diseases had earlier onset of irAEs (after median 5.4 months) than the others (after median 13 months).	[78]
NSCLC patients treated in the USA with pre-existing autoimmune conditions receiving anti-PD-1 mAbs.	2018	Retrospective observationalmulticenter	56	25	- A total of 23% of patients experienced a flare of pre-existing condition; 4 patients required systemic steroids; - A total of 38% of patients experienced new irAEs of any grade (74% were G1 or G2); eight patients required systemic steroids; - Overall, ICIs were stopped in eight patients.	[70]
Melanoma patients with pre-existing autoimmune conditions receiving anti-PD-1 mAbs.	2017	Retrospective observationalmulticenter	52	27	- A total of 38% of patients experienced a flare of pre-existing conditions requiring immunosuppressive drugs (including seven out of thirteen patients with rheumatoid arthritis); - A total of 29% of patients experienced new irAEs of any grade; - ICIs were stopped in four patients.	[76]
Melanoma patients with pre-existing autoimmune conditions receiving ipilimumab.	2016	Retrospective observationalmulticenter	30	10	- A total of 27% of patients experienced a flare of pre-existing conditions and were treated with steroids; - A total of 33% of patients experienced new irAEs G3–G5 treated with steroids or infliximab.	[71]

* For all population with pre-existing autoimmunity unless specified otherwise. Abbreviations (in alphabetic order): CTCAE—Common Terminology Criteria for Adverse Events, G—grade, ICIs—immune checkpoint inhibitors, irAEs—immune-related adverse events, mAbs—monoclonal antibodies, N/A—not applicable, No—number, NSCLC—non-small cell lung cancer, OS—overall survival, ORR—objective response rate, PD-1—programmed death-1, PFS—progression-free survival, Ref—references.

### 7.2. Patients with Pre-Existing Autoimmune Disorders Treated with Immune Checkpoint Inhibitors Likely Have a Similar Risk of All Immune-Related Adverse Events to the General Population

In numerous real-world studies, the incidence of irAEs experienced by ICIs-treated cancer patients with variable pre-existing autoimmune disorders was similar to that obtained in pivotal clinical trials, where autoimmunity was an exclusion criterion [69,70]. Taking into account the studies mentioned above (Table 3), in general, in one study (n = 10 patients with rheumatic disease), there was a higher risk of irAEs in this population compared to the general population [66]. On the other hand, in one systematic review (n = 68 patients with rheumatic disease), there were no differences in irAEs between patients with either the active or inactive condition [68]. Altogether, the low number of analyzed patients with pre-existing rheumatic disorders does not allow for a reliable answer to whether the risk of rheumatic irAEs in that group is higher.

### 7.3. Targeted Treatments Managing Immune-Related Adverse Events Do Not Compromise Immunotherapy Efficacy

As immunosuppressive agents are used in the treatment of irAEs, there is a potential risk of compromising the efficacy of ICIs in cancer. However, the relationship between ICIs and immunosuppressants remains unclear. Some studies demonstrated a favorable outcome in patients despite the administration of systemic steroids [79]. On the other hand, as mentioned in Table 3, some patients (n = 74 in two studies) had poorer survival after immunosuppressive treatment at ICIs initiation [65,69]. An interesting concept is the combination of nivolumab and ipilimumab with immunosuppressants as a prophylaxis of irAEs. Currently, this strategy is under evaluation in clinical trials, as described below [80,81,82].

In a retrospective study of patients with arthritis, tumor necrosis factor (TNF)α inhibitors administered from 3 to 15 months did not induce tumor progression [36]. Interestingly, in preclinical models of mouse melanoma, anti-TNFα antibodies increased CD8 tumor-infiltrating lymphocytes, which might suggest a favorable prognosis [83]. Additionally, in the mice models, the neutralization of TNFα improved the effectiveness of cancer immunotherapy, ameliorated immunotherapy-induced colitis, and overcame resistance to anti-PD-1 blockade in mouse melanoma [83,84,85]. This phenomenon may suggest a synergistic effect of anti-PD-1 and anti-TNFα agents. Recently, a clinical trial phase 1b was performed in patients with advanced melanoma with a combination of anti-PD-1, anti-CTLA-4, and anti-TNFα (certolizumab and infliximab), which revealed a clinical benefit in 66.7% of the patients on certolizumab and 50% in the infliximab cohort. However, this gain was occupied by hepatobiliary toxicity in 50% of the enrolled individuals [82]. Certolizumab was also combined with chemotherapy in advanced lung adenocarcinoma with 56% of partial responses in a phase-1 clinical trial [80].

Immunosuppressive agents directed against IL-6 (tocilizumab and sarilumab) in the treatment of irAEs were not evaluated in terms of their influence on tumor progression. However, a chronically elevated level of IL-6 promotes tumor cell survival, is a poor prognostic factor, and may indicate the occurrence of irAEs during ICIs treatment [86,87]. Despite this evidence, IL-6 inhibitors did not improve outcomes in various types of cancer (myeloid myeloma, prostate cancer, and renal cell carcinoma), possibly due to the high heterogeneity of cancer cells—some of them are IL-6-sensitive, but others are stimulated by additional growth factors [88,89], or a higher IL-6 level is a biomarker but not a reason for a worse prognosis. Furthermore, the signatures of the IL-6 gene were found to be increased in patients who did not respond to immunotherapy [90]. According to a preliminary study conducted in 31 patients, adding an IL-6 inhibitor to treat irAEs increased the overall response from 57.7% to 65.4% [90]. This preclinical finding warrants an ongoing phase-2 clinical trial of a combination of ICIs with an antibody against IL-6 (ipilimumab + nivolumab and tocilizumab) for the treatment of advanced melanoma, non-small cell lung cancer, and urothelial carcinoma [81].

In conclusion, it seems that immunosuppressive agents used mutually with ICIs do not compromise their efficacy but even enhance it at the cytokine level, thus reducing toxicity. Although there is a rationale in preclinical models, the results of ongoing clinical trials [81,82] are anticipated to verify this hypothesis.

## 8. Surprisingly, Patients Who Develop Immune-Related Adverse Events on Immune Checkpoint Inhibitors Have Better Cancer Clinical Outcomes

There is a rationale for considering irAEs as a predictive factor for oncological responses. To date, several studies have revealed a correlation between irAEs and better survival [79,91,92,93]. In our real-life report, melanoma patients treated with combined immunotherapy who experienced irAEs had an 80% lower risk of death (hazard ratio (HR) 0.2, 95% CI 0.07–0.57, *p* = 0.001) and disease progression (HR 0.2, 95% CI 0.09–0.47, *p* < 0.0001) [93]. In another major study of 1747 patients with metastatic or locally advanced urothelial cancer treated with anti-PD-1/PD-L1 antibodies, irAEs were reported in 28% of responders vs. 12% of non-responders [92]. Interestingly, the presence of irAEs and a specific type of toxicity could be of clinical importance. For example, cutaneous irAEs and vitiligo have been shown to be associated with overall survival (OS) benefits (HR 0.45, CI: 0.251–0.766 and HR 0.22, CI: 0.025–0.806, respectively) in melanoma patients treated with nivolumab with no significant differences in survival between other irAEs (endocrinopathies, colitis, or pneumonitis) [79]. It is noteworthy that rheumatic irAEs were not recorded in this study. However, Liew et al. [94] demonstrated that non-cutaneous irAEs were associated with a good clinical response (relative risk (RR) 2.23, CI: 1.45–3.42), especially rheumatic irAEs (RR 11.16, CI: 2.65–46.98). A possible explanation of this phenomenon is that rheumatic irAEs occur later during treatment (after a median time of 71 days compared to skin and gastrointestinal toxicities detected after 42 days [5]), and individuals who do not respond to treatment have no time to develop rheumatic irAEs. However, in the study mentioned above, this association remained in the subgroup of patients treated for more than 12 weeks. In another study, Buder-Bakhaya et al. [95] reported that patients with arthralgia have a better OS and progression-free survival (PFS) than those who do not develop this symptom (not reached vs. 17.8 months for OS and not reached vs. 4.2 months for PFS). Therefore, it seems crucial for oncologists to avoid stopping ICIs immediately after rheumatic irAEs occur, since prompt rheumatologic therapy may control irAEs, thus enabling the successful continuation of anticancer treatment.

## 9. Conclusions

Rheumatic irAEs are more frequent in clinical practice than are reported in clinical trials. Although the treatment of irAEs is similar regardless of their clinical presentation, as steroids are the mainstay, rheumatic irAEs are a distinct entity. They are long-lasting, usually persist after immunotherapy cessation, and may occur several years after stopping ICIs. Their treatment is also non-steroid-based, as patients should receive NSAIDs and DMARDs, including biologics, which seem to be safer when added to the used cancer therapy. Stable patients with pre-existing rheumatic autoimmune diseases may receive ICIs, although strict vigilance is required since they are prone to develop frequent flares (even in 40% of cases). In turn, those with non-rheumatic autoimmune diseases are usually stable when on ICIs. The incidence of irAEs is similar in patients with and without pre-existing rheumatic autoimmune diseases, although data on this issue are minimal. Antirheumatic drugs are safe to use during ICIs treatment, but it is suggested to replace non-selective immunosuppressants with antigen-directed agents. However, this recommendation is not supported by prospective studies and requires a case-by-case discussion.

## Figures and Tables

**Figure 1 ijerph-20-04926-f001:**
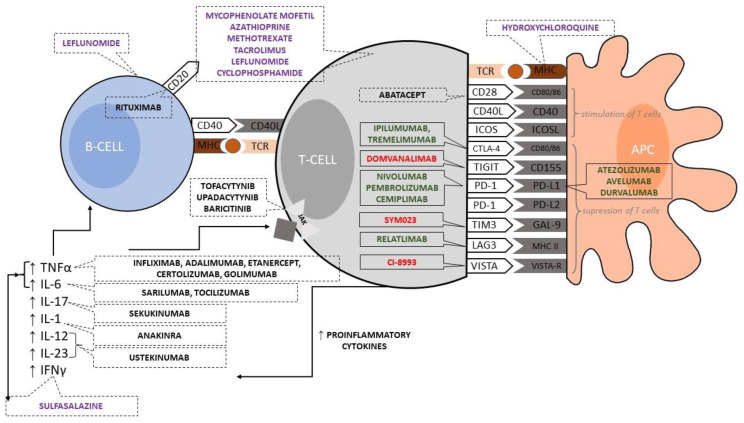
Immunological synapse with stimulatory/inhibitory receptors and interactions between immune cells. Checkpoint inhibitory agents (in a solid line frame) registered by European Medicines Agency and/or U.S. Food and Drug Administration are presented (in green color) with examples of molecules that are currently under evaluation in clinical trials (in red color). Two signals are required to induce T-cell response: (1) presentation of the antigen bound to major histocompatibility complex (MHC) on the antigen-presenting cell (APC) surface followed by its recognition by the specific T-cell receptor (TCR) (left-hand side), and (2) cluster of differentiation (CD) 28 and CD80/CD86 interaction—second stimulating signal (right-hand side). Ligands on APCs (stimulatory/inhibitor receptors) determine the response of T cells. With immune checkpoint inhibitors, activated T cells release inflammatory cytokines that act against malignant cells and can cause immune-related adverse events as an off-target effect. The presented mechanisms of immunosuppressive agents (in a dotted line frame) are simplified. Classic immunosuppressive/immunomodulatory agents are in violet. New medications with more precise, tailored action modes, such as biologics or tyrosine kinase inhibitors, are in black. They are registered in rheumatic diseases but can also be used to manage rheumatic immunologic-related adverse events of immunologic checkpoint inhibitors in oncology. Abbreviations: APC—antigen-presenting cell, CD—cluster of differentiation, CTLA-4—cytotoxic T-lymphocyte antigen 4, GAL9—galectin 9, ICOS/ICOSL—inducible T-cell co-stimulator/ligand, IFN—interferon, IL—interleukin, LAG-3—lymphocyte activation gene 3 protein, PD-1/PD-L1 programmed cell death protein 1/ligand, TIGIT—T-cell immunoreceptor with immunoglobulin and ITIM domains, TIM-3—T-cell immunoglobulin mucin receptor 3, VISTA/VISTA-R—V-domain Ig suppressor of T-cell activation/receptor, TCR—T-cell receptors, TNFα—tumor necrosis factor alpha.

**Table 1 ijerph-20-04926-t001:** Comparison of ‘oncological’ Common Terminology Criteria for Adverse Events and Rheumatology Common Toxicity Criteria.

	Common Terminology Criteria for Adverse Events Version 5.0 *	Rheumatology Common Toxicity Criteria Version 2.0 **
Grade 1—mild	Asymptomatic or mild symptoms, clinical or diagnostic observations only, intervention not indicated.	Asymptomatic, or transient Short duration (<1 week) No change in lifestyle No medication or over-the-counter
Grade 2—moderate	Minimal, local, or noninvasive intervention indicated, limiting age-appropriate instrumental activities of daily living (preparing meals, shopping for groceries or clothes, using the telephone, managing money, etc.)	Symptomatic Duration 1–2 weeks Alter lifestyle occasionally Medications give relief
Grade 3—severe	Hospitalization or prolongation of hospitalization indicated, disabling, limiting self-care activities of daily living (bathing, dressing and undressing, feeding self, using the toilet, taking medications, and not bedridden.)	Prolonged symptoms, reversible Major functional impairment Prescription medications/partial relief; hospitalized < 24 h Temporary or permanent study drug discontinuation
Grade 4—life-threatening	Life-threatening consequences, urgent intervention indicated.	At risk of death Substantial disability, especially if permanent Hospitalized > 24 h Permanent study drug discontinuation
Grade 5—death		

Table 1 references: *—[33]; **—[34].

## Data Availability

Not applicable.

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
