# Peer review of "Immune Checkpoint Inhibitor Related Rheumatological Complications: Cooperation between Rheumatologists and Oncologists"

_ijerph, 2023, doi:10.3390/ijerph20064926_

Round 1

Reviewer 1 Report

This is a very interesting Review article on immune checkpoint inhibitor-related rheumatological complications which highlights the importance of a thorough interdisciplinary cooperation between oncologists and rheumatologists in the management of these clinical cases.

The contents are exhaustively described.

Minor suggestions:

- Please add a brief Methods section highlighting the search strategy (including databases and keywords employed).

- Please embedd relevant references in Tables 1 and 2 (avoid citing references in the table heading).

- I suggest to extend the conclusions (paragraph 8). Moreover, I would avoid structuring paragraph 8 as bullet points.

Author Response

  1. This is a very interesting Review article on immune checkpoint inhibitor-related rheumatological complications which highlights the importance of thorough interdisciplinary cooperation between oncologists and rheumatologists in the management of these clinical cases. The contents are exhaustively described

Answer:  Thank you very much for this comment and appreciation of our work.

  1. Please add a brief Methods section highlighting the search strategy (including databases and keywords employed).

Answer: The manuscript has been improved accordingly (Methods section).

  1. Please embed relevant references in Tables 1 and 2 (avoid citing references in the table heading).

Answer: We improved the manuscript according to the Reviewer’s suggestion.

  1. I suggest to extend the conclusions (paragraph 8). Moreover, I would avoid structuring paragraph 8 as bullet points.

Answer: Thank you for that remark. The manuscript has been corrected accordingly.

Author Response

  1. Authors discussed the topic “Immune checkpoint inhibitors-related rheumatological complications: cooperation between rheumatologists and oncologists”. Although the final conclusion is theoretical but still this manuscript paves the way for the further research in the above-mentioned area. The manuscript is nicely written and proper citation to the retrospective studies was given. Methodology adopted by the authors is scientifically justified. However, few points need to be considered before accepting the manuscript:

Answer: The authors would like to acknowledge Reviewer#2 contribution in peer-reviewing of our manuscript and valuable comments. 

  1. Role of genes is not discussed anywhere in the manuscript. Some patients may develop or experience rheumatic conditions sooner than other patients depending on the genomic difference and genome based physiologic difference.

Answer: Thank you for this inspiring remark. We addressed this issue in the new version of the manuscript - Section 4.

  1. Point 7 “Surprisingly, patients who develop immune-related adverse events on immune checkpoint inhibitors have better cancer clinical outcomes”, please add more about the better clinical outcome for cancer.

Answer: We provided more details on that issue in section 8 of the revised version of the manuscript.

  1. Please add discussion part.

Answer: Our paper is not an original study, so we did not provide the specific ‘discussion’ section. Instead, parts corresponding to the discussion are incorporated into the subsections. We hope that this format is acceptable to the Reviewer.

  1. Although the authors present the manuscript very well, the conclusion part can be further improved.

Answer: The manuscript has been improved accordingly.